# Carbonate chemistry and carbon sequestration driven by inorganic carbon outwelling from mangroves and saltmarshes

Gloria M. S. Reithmaier [1] ✉, Alex Cabral [1], Anirban Akhand[2,3], Matthew J. Bogard [4], Alberto V. Borges [5], Steven Bouillon[6], David J. Burdige [7], Mitchel Call[8], Nengwang Chen[9], Xiaogang Chen [10], Luiz C. Cotovicz Jr[11,12], Meagan J. Eagle[13], Erik Kristensen [14], Kevin D. Kroeger[13], Zeyang Lu[15], Damien T. Maher [8], J. Lucas Pérez-Lloréns [16], Raghab Ray[17], Pierre Taillardat[18], Joseph J. Tamborski[7], Rob C. Upstill-Goddard[19], Faming Wang [20], Zhaohui Aleck Wang[21], Kai Xiao[22], Yvonne Y. Y. Yau[1] & Isaac R. Santos [1]

Mangroves and saltmarshes are biogeochemical hotspots storing carbon in sediments and in the ocean following lateral carbon export (outwelling). Coastal seawater pH is modified by both uptake of anthropogenic carbon dioxide and natural biogeochemical processes, e.g., wetland inputs. Here, we investigate how mangroves and saltmarshes influence coastal carbonate chemistry and quantify the contribution of alkalinity and dissolved inorganic carbon (DIC) outwelling to blue carbon budgets. Observations from 45 mangroves and 16 saltmarshes worldwide revealed that >70% of intertidal wetlands export more DIC than alkalinity, potentially decreasing the pH of coastal waters. Porewater-derived DIC outwelling ($81 \pm 47$ mmol m$^{-2}$ d$^{-1}$ in mangroves and $57 \pm 104$ mmol m$^{-2}$ d$^{-1}$ in saltmarshes) was the major term in blue carbon budgets. However, substantial amounts of fixed carbon remain unaccounted for. Concurrently, alkalinity outwelling was similar or higher than sediment carbon burial and is therefore a significant but often overlooked carbon sequestration mechanism.

The ocean is an important sink for anthropogenic carbon dioxide ($CO_2$) emissions. Increased $CO_2$ dissolution in the ocean causes ocean acidification and threatens a wide range of marine organisms and ecosystems[1]. Coastal ecosystems are particularly vulnerable to ocean acidification[2], which jeopardizes their ecosystem services and functions, i.e., habitat for biodiversity, fisheries, coastal protection, and tourism[3]. Processes impacting coastal water acidification are more complex than in the open ocean resulting in larger spatial and temporal variability of the carbonate system[4]. In addition to the dissolution of anthropogenic $CO_2$ into seawater, the coastal carbonate system can be impacted by many local carbon sources, such as upwelling, groundwater, riverine, and wetland inputs[5].

Intertidal wetlands, e.g., mangroves and saltmarshes, are biogeochemical hotspots storing large amounts of carbon (54 Tg C y$^{-1}$) in their sediments[6]. They also laterally export carbon to the coastal ocean (termed outwelling)[7–10], which alters carbonate chemistry and thus impacts seawater pH. Mangroves and saltmarshes produce organic carbon that is partially mineralized, releasing inorganic carbon in the form of carbonate alkalinity (mostly as $HCO_3^-$ at pH < 8) and dissolved inorganic carbon (DIC = $CO_2 + HCO_3^- + CO_3^{2-}$). Exported total alkalinity (TA) represents a long-term carbon sink[11] and can

buffer the coastal ocean against acidification. Exported DIC, in contrast, can enhance coastal acidification by forming carbonic acid when $CO_2$ reacts with water and partly returns $CO_2$ to the atmosphere via air–sea exchange.

Different diagenetic processes produce TA and/or DIC in wetland sediments. Aerobic respiration produces mostly DIC, whereas anaerobic respiration (denitrification, sulfate reduction, manganese reduction, and iron reduction) produces both DIC and TA[12]. TA produced during anaerobic respiration only contributes to a permanent TA increase if respiration is coupled with the removal of reduced compounds, i.e., sulfide precipitated as pyrite or nitrogen outgassing ($N_2$) by denitrification[13,14]. Porewater acidification following $CO_2$ production from organic matter degradation can drive metabolic carbonate dissolution, which also produces TA[15]. Drainage of intertidal wetland sediments by tidal pumping (porewater export driven by tidal variations in hydraulic gradients) and bio-irrigation (porewater export driven by benthic organisms) transports DIC- and TA-enriched porewater to surface waters and eventually to the coastal ocean[16–18].

Examining TA and DIC in porewater, surface water, and during outwelling gives valuable insights into how intertidal wetlands affect the carbonate system of coastal waters. The TA:DIC ratio relates to buffer factors and is thus a major property of carbonate chemistry, determining the buffering capacity of seawater against external acid inputs[19]. Consequently, TA and DIC inputs drive pH changes and influence the capacity of seawater to take up anthropogenic $CO_2$, affecting ocean acidification and carbon sequestration.

Here, we investigate whether mangroves and saltmarshes buffer coastal waters against acidification and re-examine their potential to sequester atmospheric $CO_2$. We compiled TA and DIC contents in porewater and surface water (measured during time-series and spatial surveys) at mangrove- and saltmarsh-dominated systems worldwide (Fig. 1)[20,21]. We also upscaled compiled TA and DIC outwelling rates globally to evaluate whether intertidal wetlands export more TA or DIC and to update mangrove and saltmarsh carbon budgets.

## Results and discussion

### Intertidal wetlands produce more DIC than alkalinity

Mangrove and saltmarsh sediments are major sources of TA and DIC to surrounding waters. TA (153–34,500 µmol kg⁻¹) and DIC (844–28,200 µmol kg⁻¹) in porewaters were two- to three times higher than in surface waters (5–11,500 and 37–9390 µmol kg⁻¹, respectively, Supplementary Fig. 1). Slopes of TA:DIC regressions per site, normalized to the median salinity of each site, were $0.82 \pm 0.07$ (0.78) (median ± SE (average)) in porewater and $0.75 \pm 0.04$ (0.70) in surface water (Supplementary Table 2), being slightly higher in saltmarshes than in mangroves. Multiple processes produce TA and DIC within sediments at different ratios resulting in specific TA:DIC slopes: aerobic respiration (−0.2), denitrification (0.8), sulfate reduction (1), metabolic carbonate dissolution (2), manganese reduction (4), and iron reduction (8)[12]. The observed porewater and surface water slopes imply a combination of TA and DIC production during aerobic respiration, sulfate reduction, and at some sites, denitrification and metabolic calcium carbonate dissolution[22–24]. 38% of mangrove and 35% of saltmarsh surface water samples exceeded typical seawater values of 2350 µmol kg⁻¹ [11], suggesting a net production of TA after accounting for TA consumption by biogeochemical processes such as sulfide oxidation and nitrification.

Most intertidal wetlands, however, produced more DIC than TA. Averaging values per site, porewater TA:DIC ratios ($0.90 \pm 0.04$ (0.90)) were slightly lower than surface water ratios ($1.03 \pm 0.01$ (1.03)) (Fig. 2a, b). The use of TA:DIC ratios as a proxy for acidification or buffering is explained in detail in the Supplementary Methods. Surface waters surrounding intertidal wetlands often receive some freshwater (TA:DIC ratio ≈1.0 varies amongst catchments) or seawater (TA:DIC ratio ≈1.1) inputs[4]. Surface water TA:DIC ratios increased with salinity and dissolved oxygen (Supplementary Fig. 2), which are typically higher in seawater than within intertidal wetlands[25,26]. Therefore, TA:DIC ratios exceeding 1 in surface waters at most sites are partially due to mixing with seawater during flood tides. TA:DIC ratios were negatively correlated with the natural porewater tracer radon (Supplementary Fig. 3), suggesting that inputs of porewater enhance acidification. In porewaters, TA:DIC ratios increased with dissolved

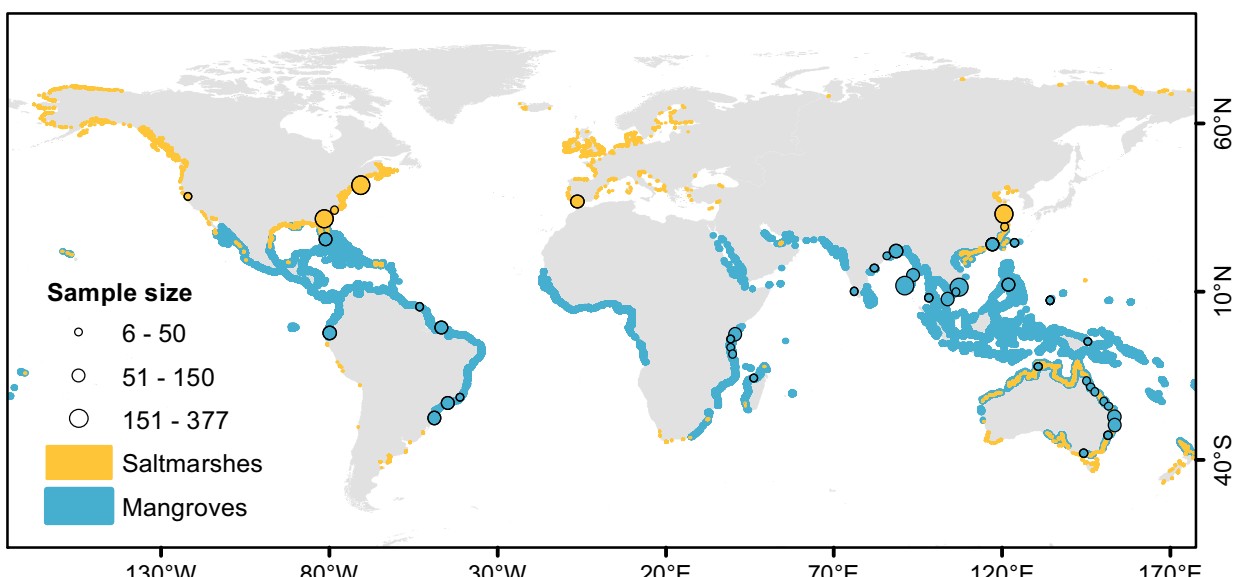

**Fig. 1 | Location of mangroves and saltmarshes with alkalinity (TA) and dissolved inorganic carbon (DIC) observations.** TA and DIC were measured in porewater and surface water at 38 mangrove- and 8 saltmarsh-dominated creeks and estuaries. The location of some sites was slightly adjusted to allow visualization (precise coordinates are available on Supplementary Table 1). The underlying global mangrove (blue areas)[20] and saltmarsh (orange areas)[21] distributions were retrieved from existing databases. The ecosystems overlap in many coastlines, such as Australia, China, and the Gulf of Mexico. The sample size scale refers to the number of measurements at each site.

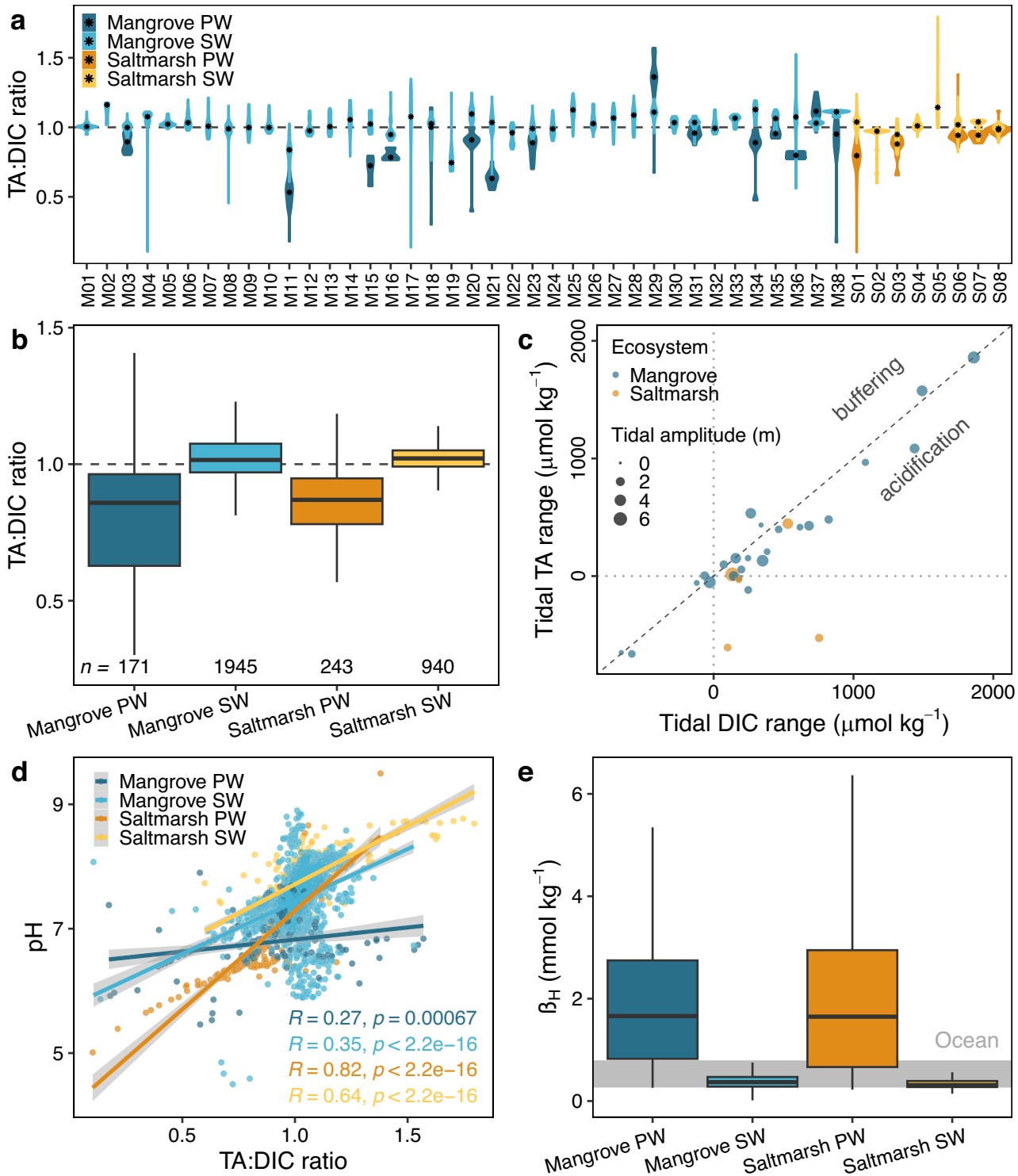

**Fig. 2 | Inorganic carbon speciation in intertidal wetlands impacts coastal seawater pH.** TA:DIC ratios **a** in porewater (PW) and surface water (SW) at each site, asterisks represent median values and **b** distribution per group of samples. **c** Tidal alkalinity (TA) and dissolved inorganic carbon (DIC) ranges estimated from the difference in concentrations between low and high tide. A large tidal range implies a large porewater source. Scales were fit to improve the readability excluding 1 outlier. **d** Regressions between pH and TA:DIC ratios. Gray areas indicate the 95% confidence intervals of the regressions. **e** Buffer factor $\beta_H$ per group and ocean range[19] presented as a grey band. Boxplots in Fig. 2b, d indicate the median (middle line), 25th, 75th percentile (box) and 5th and 95th percentile (whiskers). Sources of datasets are listed in Supplementary Table 1.

organic carbon (DOC) (Supplementary Fig. 2), potentially due to organic alkalinity contributions or a higher contribution of anaerobic respiration to the total respiration due to higher organic matter loading[27,28].

At 61% of the sites, TA and DIC in surface waters measured during timeseries were higher at low tide than at high tide (Fig. 2c). Therefore, TA and DIC are likely derived from tidally-driven porewater export, which is greatest during low tides. Systems with greater tidal

amplitudes had high tidal TA and DIC ranges with higher contents at low tide (i.e., minimum water level). Only some systems with small tidal amplitudes (<2 m), had slightly higher TA or DIC at high than at low tide. At most sites (76%), DIC differences between low and high tide were larger than TA differences, indicating that more DIC was exported by tidal pumping (Fig. 2c). As a result, surface water pH decreased by 0.3 ± 0.1 (0.4) units during low tides compared to high tides (Supplementary Table 2). The relatively larger DIC production and export to surface waters is supported by larger estuarine DIC than TA inputs at four out of six mangrove-dominated estuaries, as revealed by mixing models (Supplementary Figs. 4 and 5).

Overall, tidal dynamics and mixing models reveal that 23 out of 33 mangrove sites and all saltmarsh sites (n = 10) had higher DIC than TA inputs, potentially enhancing local acidification. Sites without water level data were excluded from the analysis. The local pH was positively correlated with TA:DIC ratios, however, data were scattered around TA:DIC ratios of ≈1 (Fig. 2d), where seawater reaches a minimum buffering capacity. This minimum buffering likely causes larger variations in seawater pH for a given acid-base perturbation[9,28].

The buffer factor $\beta_H$, which quantifies the resistance to pH changes at the addition of an acid or base, was five times higher in porewater than in surface waters. Furthermore, $\beta_H$ in porewater was two- to five times higher than in oceanic waters[19] (Fig. 2e). This emphasizes the high buffer capacity of mangrove and saltmarsh porewaters. Earlier compilations of buffer factors in the marine systems have focused on coral reefs[29] and the open ocean[19]. The spatial and temporal variability of buffer factor $\beta_H$ in mangroves and saltmarshes exceed other marine ecosystems. Therefore, the impact of the acid release from wetland porewaters on the pH varies considerably. The apparent paradox of increasing buffer capacity despite increasing acidification in intertidal wetlands can be explained by seawater reaching a minimum buffering capacity when TA = DIC at pH ≈7.5. Most carbonate ions become bicarbonate ions when the halfway point between dissociation constants of carbonic acid is reached[19]. When TA:DIC ratios further decrease after this minimum (<1), the water buffering capacity increases with increasing bicarbonate content[9]. Consequently, the increased buffer capacity of porewaters minimizes the acidification potential caused by $CO_2$ release from intertidal wetlands.

The majority of intertidal wetlands produce and export more DIC than TA and thus likely acidify surrounding waters. However,

acidification by intertidal wetlands is highly site-specific, impacted by external nutrient inputs, mixing with nearshore waters, and varies seasonally (Supplementary Figs. 3 and 6). Future studies should venture into continental shelf waters to examine the extent of wetland-driven acidification on marine biogeochemistry and allow for comparison with anthropogenic-driven ocean acidification.

## Inorganic carbon outwelling a major carbon fate

We compiled TA and DIC outwelling fluxes from mangroves and saltmarshes to update the carbon budgets of intertidal wetlands (Supplementary Table 3). Most sites (71%) exported more DIC than TA to the coastal ocean (Supplementary Fig. 7), which is consistent with observations of TA and DIC production (Fig. 2). The TA:DIC outwelling ratios (median ± SE (average): 0.8 ± 0.2 (1.0)) ranged from 0.1 to 4 in mangroves and from 0.6 to 1 in saltmarshes. This indicates that TA and DIC originating from sediments are exported to adjacent waters, affecting the carbonate chemistry and potentially enhancing coastal acidification. Elevated inorganic carbon and $CO_2$ outgassing was associated with intertidal wetland outwelling on continental shelves in Japan[30], Brazil[31], and the US[32,33]. In addition to inorganic carbon, organic carbon outwelling can also enhance coastal acidification and $CO_2$ outgassing due to enhanced microbial $CO_2$ production[34]. These observations suggest that intertidal wetlands might modify the pH of nearshore shelf waters, but the magnitude and scale of the impact on seawater pH are highly site-specific, depending on the climate, geomorphology, hydrology, and size of the system.

On a local scale, TA and DIC outwelling varied over tidal cycles[35,36], across seasons[9,37], in response to episodic weather events[38,39], and was controlled by carbonate dissolution in some coastlines such as the Everglades[18] (Supplementary Table 3). DIC outwelling from mangroves was approximately two times higher during wet than dry seasons[37,38,40,41]. Macrotidal Chinese saltmarshes[36,42–45] had DIC outwelling rates exceeding those in microtidal US saltmarshes[9,32,34] by a factor of 20 (Fig. 3). However, on a global scale, temperature, precipitation, tidal amplitude, sediment, and carbon accumulation rates had minor or undetectable influences on outwelling (Supplementary Figs. 8–10, Supplementary Table 4). Several methods have been used to quantify outwelling, and the intertidal area associated with outwelling is often prone to large uncertainties[46]. Since it is challenging to estimate outwelling, most rates are based on two tidal cycles. Only 17

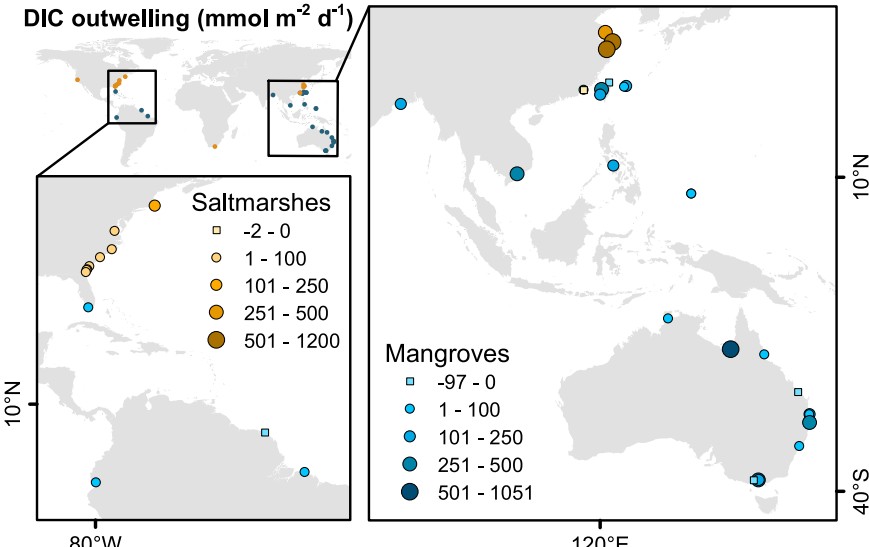

**Fig. 3 | Most studies measuring dissolved inorganic carbon (DIC) outwelling rates from intertidal wetlands were conducted in the USA, China, and Australia.** Rates are scaled to the intertidal wetland area. The location of some sites was adjusted to allow visualization. The precise coordinates are available on Supplementary Table 3.

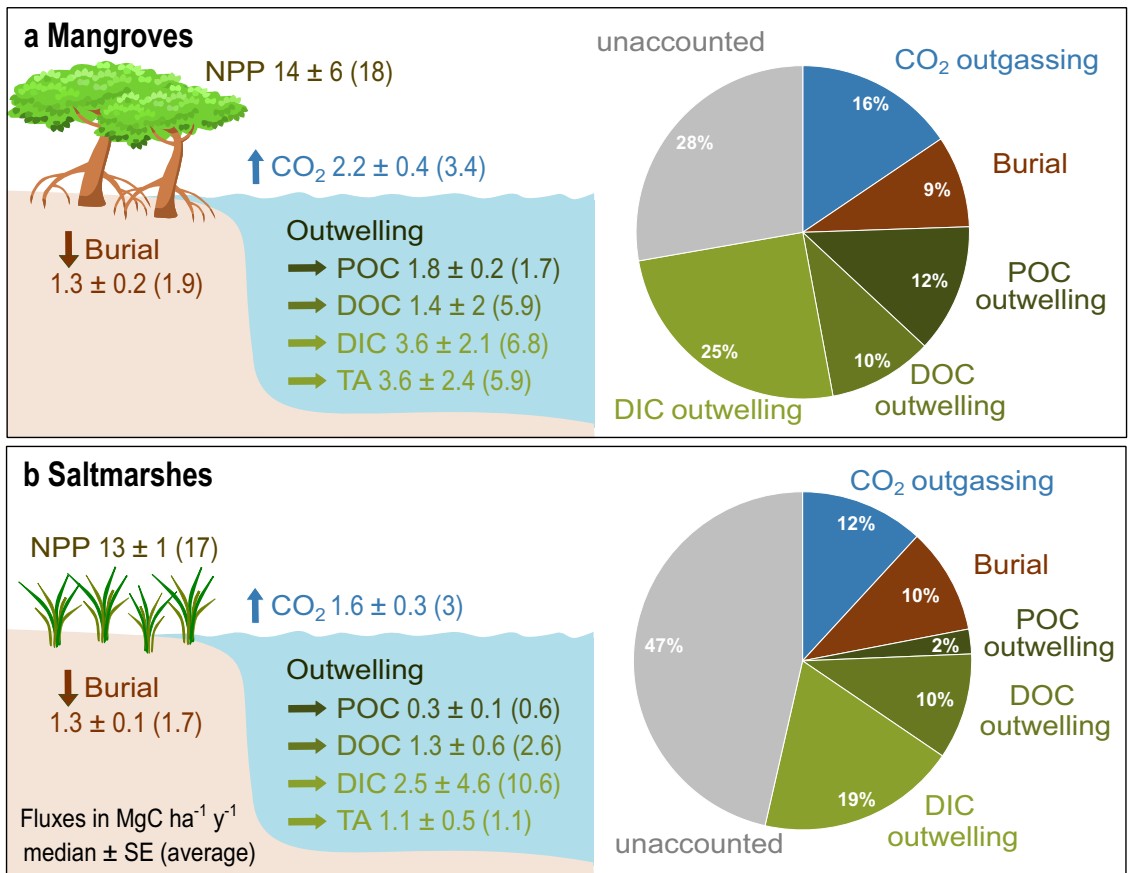

**Fig. 4 | Fates of mangrove and saltmarsh production are dominated by inorganic carbon export.** However, **a** mangrove and **b** saltmarsh carbon budgets (net primary production (NPP) minus major carbon fates) remain unbalanced (Supplementary Table 5). Pie charts are based on median values. Using averages would close carbon budgets, but averages are unlikely to be representative of the skewed dataset with clear outliers. The values for burial rates[6], aquatic $CO_2$ outgassing, dissolved organic carbon (DOC), and particulate organic carbon (POC) outwelling rates[47] were retrieved from the literature. Our new dissolved inorganic carbon (DIC) outwelling estimates are based on Supplementary Table 4.

out of 40 sites have some seasonal data. Larger, seasonal datasets covering a broad global distribution may be required to establish robust relationships on a global scale.

The global area-weighted DIC outwelling in mangroves (median ± SE (average): 81 ± 47 (155), range: −97 to 1051 mmol m$^{-2}$ d$^{-1}$; $n = 26$) and saltmarshes (57 ± 104 (242), −2 to 1200 mmol m$^{-2}$ d$^{-1}$; $n = 14$), was the dominant fate of carbon fixed by net primary production, exceeding carbon burial, aquatic $CO_2$ outgassing, particulate and dissolved organic carbon outwelling (Fig. 4)[6,47]. In spite of a large natural variability, these global-scale estimates are consistent with a series of recent local-scale carbon budgets demonstrating that DIC outwelling exceeded carbon burial in mangroves of Australia[38,48], and saltmarshes of the US[9] and China[26,49]. However, even when accounting for DIC outwelling as an additional carbon output, the fate of 28% and 47% of the carbon fixed by net primary production (NPP) remain unidentified for mangroves and saltmarshes, respectively[6,47] (Fig. 4 and Supplementary Table 5). $CO_2$ outgassing from exposed mangrove sediments, which is facilitated by biogenic structures such as pneumatophores and crab burrows, could be another major carbon fate[50], but current datasets remain small.

While DIC outwelling can be followed by $CO_2$ outgassing and thus return carbon back to the atmosphere, exported TA represents a long-term carbon sink since it stays dissolved in the ocean for millennia[11]. TA outwelling rates were threefold higher in mangroves (81 ± 55 (134), −1 to 951 mmol m$^{-2}$ d$^{-1}$; $n = 17$) and slightly smaller in saltmarshes (25 ± 11 (26), −2 to 69 mmol m$^{-2}$ d$^{-1}$; $n = 6$) compared to the global average of carbon burial of these intertidal wetlands[6]. Recognizing lateral TA

exports as a carbon sequestration mechanism will enhance the perceived value of those blue carbon ecosystems[11] and give insights into buffering and acidifying processes in adjacent coastal waters. Additional seasonal studies in all parts of the world are essential to reduce uncertainty and refine budgets.

## Implications and perspectives

When upscaling median values to the global area of mangroves (140,000 km$^2$)[51] and saltmarshes (55,000 km$^2$)[21], intertidal wetlands export 5.3 ± 3.2 Tmol y$^{-1}$ DIC and 4.6 ± 2.8 Tmol y$^{-1}$ TA to the coastal ocean. DIC exports by global rivers to the ocean are estimated at 32 Tmol y$^{-1}$, only about six times greater than our estimate of outwelling from mangroves and saltmarshes combined[11]. The global alkalinity balance of the ocean, which is dominated by riverine input and calcium carbonate burial, does not currently include inputs by intertidal wetlands[11]. Our results suggest that TA outwelling from coastal wetlands is equivalent to 7% of the total TA sources (71 Tmol y$^{-1}$) into the ocean, exceeding other TA sources such as submarine groundwater discharge (1.0 Tmol y$^{-1}$), denitrification (1.5 Tmol y$^{-1}$), submarine silicate weathering (2.8 Tmol y$^{-1}$), and organic matter burial (3.0 Tmol y$^{-1}$) (Supplementary Table 6). Hence, despite their small area, intertidal wetlands are global hotspots for carbon and TA production and outwelling and should be accounted for in global marine carbon budgets.

To refine global TA and DIC outwelling rate estimates from intertidal wetlands, which will impact coastal acidification and carbon budgets, more studies are required in South America, Africa, Europe, and Southeast Asia. Studies on inorganic carbon dynamics in

saltmarshes should cover a broader global distribution, whereas research in mangroves should be conducted in different seasons over complete spring-neap tidal cycles. Methodological differences, spatial scales, geomorphological settings, seasonal changes, and episodic weather events should be considered to refine global upscaling.

Overall, our compilation of TA and DIC contents and outwelling rates in mangroves and saltmarshes revealed greater DIC than TA exports at most sites, likely accelerating coastal acidification. The decrease of coastal pH due to porewater input is partially hampered by the high buffer capacity of porewaters compared to surface waters. The effect of acidification is likely highest near intertidal wetlands, potentially affecting the carbonate system of estuaries, coral reefs, and nearshore coastal areas. TA production is important not only from an ocean acidification context but also from a carbon budget perspective. The large TA exports represent an overlooked carbon sequestration mechanism that more than doubles the perceived role of mangroves and saltmarshes via sediment carbon burial.

## Methods

### Data compilation

Contents of TA and DIC in porewater ($n = 414$) as well as in surface water ($n = 2885$) measured during timeseries (fixed location) and spatial surveys (fixed time period) were compiled from 38 mangrove- and 8 saltmarsh-dominated creeks and estuaries (Fig. 1, Supplementary Table 1). The full dataset can be accessed at Pangea (https://doi.org/10.1594/PANGAEA.949660)[52]. We used data from creeks that were predominantly surrounded by mangrove or saltmarsh vegetation and with minimal confounding factors such as mixed vegetation or large catchments. These creeks were located in either pristine (i.e., minimally impacted) or anthropogenically impacted estuaries or coastal areas. Anthropogenically impacted areas were defined as areas that were affected by nearby urban or agricultural activities, potentially delivering pollutants, e.g., sewage or fertilizers, to creeks. We also included data from pristine mangrove- and saltmarsh-dominated estuaries. When available, environmental parameters were recorded, i.e., season, salinity, temperature, pH, dissolved oxygen, water level, porewater tracer radon, partial pressure of carbon dioxide, dissolved organic carbon, particulate organic carbon, nitrogen oxides, ammonium, total nitrogen, phosphate, and total phosphorus.

Literature values for TA ($n = 52$) and DIC ($n = 115$) outwelling at 26 mangrove and 14 saltmarsh sites were compiled (Supplementary Table 3). There was a partial intersection between the study sites where outwelling rates were measured (Fig. 3 and S8) and sites with TA and DIC observations (Fig. 1). Outwelling rates were averaged per site, and global medians ± SE (averages) were calculated for mangroves and saltmarshes (Supplementary Table 4). We retrieved site-specific parameters of outwelling sites, including tidal range, sediment accumulation rates, and carbon accumulation rates from global datasets[6]. Average annual temperature and average annual precipitation were gathered from corresponding publications or nearest weather stations.

### Calculations and modeling

The use of TA:DIC ratios as a proxy for acidification or buffering is explained in detail in the supplementary information (Supplementary Methods and Supplementary Figs. 11–13). Slopes of salinity normalized TA and DIC regressions were calculated to examine potential impacts on seawater pH and dominant biogeochemical pathways[53] (Supplementary Table 2). Salinity normalized TA and DIC were calculated using the mean salinity and, as nonzero TA or DIC endmember, the intercept of the regression between TA or DIC and salinity for each site. Determining dominant biogeochemical pathways for each site was beyond the scope of this study since it requires analysis of respiration pathways (e.g., aerobic respiration and sulfate reduction) and carbon isotopic signatures.

To test the impact of tidal variation on the carbonate chemistry, tidal ranges of DIC, TA, and pH were quantified. Tidal ranges were estimated from differences between DIC, TA, and pH values at low and at high tides. Minimum and maximum water levels were used to define low and high tides.

The standard estuarine mixing model was used to calculate estuarine TA and DIC sources/sinks in surface water along six pristine mangrove estuaries with a clear salinity gradient[54]. Conservative mixing lines were estimated from TA and DIC at the lowest and highest salinities at each site. The deviations between TA and DIC, measured during spatial surveys, and conservative mixing lines were calculated as a percentage and averaged for each estuary. Positive deviations from conservative mixing lines indicate a source within the estuary, whereas negative values indicate a sink.

The buffer factor $\beta_H$ was calculated based on carbonate parameters obtained in CO2SYS using TA, DIC, salinity, and temperature as input parameters. In CO2SYS, the dissociation constants from Millero[55], "KHSO$_4$" from Dickson[56], and the "[B]T Value" from Lee et al.[57] were chosen. $\beta_H$ was calculated based on the equations derived by Egleston et al.[19].

To examine drivers of TA:DIC ratios and outwelling rates, linear regressions and corresponding Pearson coefficients ($R$) and $p$-values were determined.

## Data availability

The biogeochemical data, which includes TA, DIC, season, salinity, temperature, pH, dissolved oxygen, water level, porewater tracer radon, partial pressure of carbon dioxide, dissolved organic carbon, particulate organic carbon, nitrogen oxides, ammonium, total nitrogen, phosphate, and total phosphorus at 38 mangrove and 8 saltmarsh sites, generated in this study have been deposited in the PANGEA database under accession code https://doi.org/10.1594/PANGAEA.949660[52]. Calculations of TA:DIC ratios, salinity normalized TA:DIC regressions, tidal ranges, and tidal pH ranges per site are provided in Supplementary Table 2. The literature summary of TA and DIC outwelling rates (including drivers, i.e., season, temperature, precipitation, tidal range, sediment accumulation rates, and carbon accumulation rates) at 26 mangrove and 14 saltmarsh sites are provided in Supplementary Tables 3 and 4.

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

## Acknowledgements

This project was funded by the Swedish Research Council (2019-03930 and 2020-00457) and received by I.R.S. We acknowledge the multiple colleagues who contributed to the data collection and analysis used in this investigation. Any use of trade, firm or product names is for descriptive purposes only and does not imply endorsement by the US Government.

## Author contributions

G.M.S.R. managed the project, processed the data, made all figures, and wrote the first draft with support from I.R.S., who also designed the project and obtained funding. A.C. performed some of the outwelling compilations. G.M.S.R., A.C., A.A., M.J.B., A.V.B., S.B., D.J.B., M.C., N.C., X.C., L.C.C., M.J.E., E.K., K.D.K., Z.L., D.T.M., J.L.P., R.R., P.T., J.J.T., R.C.U., F.W., Z.A.W., K.X., Y.Y.Y.Y., and I.R.S. provided original data, contributed to discussions, and edited the paper. All authors, except for G.M.S.R. and I.R.S., contributed equally to the paper.

## Funding

## Competing interests

The authors declare no competing interests.

## Additional information

[1]Department of Marine Sciences, University of Gothenburg, 41319 Gothenburg, Sweden. [2]Department of Ocean Science, Hong Kong University of Science and Technology, Kowloon, Hong Kong SAR, China. [3]Coastal and Estuarine Environment Research Group, Port and Airport Research Institute, 3-1-1 Nagase, Yokosuka 239-0826, Japan. [4]Department of Biological Sciences, University of Lethbridge, Lethbridge, AB, Canada. [5]Chemical Oceanography Unit, University of Liège, 4000 Liège, Belgium. [6]Department of Earth and Environmental Sciences, KU Leuven, 3001 Leuven, Belgium. [7]Department of Ocean and Earth Sciences, Old Dominion University, Norfolk, VA 23529, USA. [8]Faculty of Science and Engineering, Southern Cross University, Lismore, NSW 2480, Australia. [9]State Key Laboratory of Marine Environment Science, Xiamen University, Xiamen 361102, China. [10]School of Engineering, Westlake University, Hangzhou 310024, China. [11]Department of Marine Chemistry, Leibniz Institute for Baltic Sea Research, Warnemünde, Germany. [12]Institute of Marine Sciences (LABO-MAR), Federal University of Ceará (UFC), Fortaleza, Brazil. [13]Woods Hole Coastal and Marine Science Center, U.S. Geological Survey, 384 Woods Hole Road, Woods Hole, MA 02543, USA. [14]Department of Biology, University of Southern Denmark, 5230 Odense, Denmark. [15]College of the Environment and Ecology, Xiamen University, Xiamen 361102, China. [16]Instituto Universitario de Investigación Marina (INMAR), University of Cádiz, Puerto Real, Cádiz, Spain. [17]Atmosphere and Ocean Research Institute, The University of Tokyo, Tokyo, Japan. [18]NUS Environmental Research Institute, National University of Singapore, Singapore 117411, Singapore. [19]School of Natural and Environmental Sciences, Newcastle University, Newcastle upon Tyne NE1 7RU, UK. [20]Xiaoliang Research Station of Tropical Coastal Ecosystems, Chinese Academy of Sciences, Guangzhou 510650, China. [21]Department of Marine Chemistry and Geochemistry, Woods Hole Oceanographic Institution, Woods Hole, MA 02543, USA. [22]School of Environmental Science and Engineering, Southern University of Science and Technology, Shenzhen, China. ✉e-mail: Gloria.Reithmaier@gu.se

