## [Peer Review File · Nature Communications]

Carbonate chemistry and carbon sequestration driven by inorganic carbon outwelling from mangroves and saltmarshesREVIEWERS' COMMENTS

Reviewer #1 (Remarks to the Author):

The authors compiled and analyzed TA and DIC data in porewater and surface water from 38 mangrove- and 8 saltmarsh-dominated creeks and estuaries. The data collection and following analyses are thorough, and the presentation of the results is very detailed and comprehensible. Major findings of this study include 1) about two thirds of the studied intertidal wetlands export more DIC than TA; 2) while porewater derived DIC outwelling was the major term in blue carbon budgets, substantial amount of NPP remain uncounted for in intertidal wetlands. None of these conclusions are new findings and are well known to the community.

I wholeheartedly support the publication of this paper as the authors have done a fantastic job of assimilation the results from all these studies but the lack of any substantial new finding makes it difficult to recommend it for publication in a high impact journal.

Minor comments:

Title: title used in the manuscript is different from the title in supporting information.

Reviewer #2 (Remarks to the Author):

General Comments:

The authors present a study in which they examine DIC and TA concentrations, environmental controls, and outwelling in mangrove and saltmarsh systems. They assemble a global dataset of over 3000 porewater and surface water samples measured during timeseries and spatial surveys. The authors report that for the majority of their sites, more DIC than TA was exported to the coastal ocean, potentially enhancing coastal acidification. DIC outwelling was the primary (characterized) fate of primary production by both mangroves and saltmarshes--greater than CO₂ outgassing, burial, and organic carbon outwelling. Moreover the magnitude of DIC export from mangroves and saltmarshes combined was equal to ~15% of riverine-exported DIC, though this term is not yet accounted for in global carbon budgets.

This is a novel and interesting study that will be well received by the readers of this journal. The manuscript is well written, clear, and well organized, and I think this is a strong and comprehensive dataset. There are a lot of display items (4 figures + 13 supplemental figures and 6 supplemental tables), but they all seem to serve a purpose, so I don't recommend dropping any. The Methods are quite brief, which I realize is the norm for these short articles. However, they could include more information about the TA:DIC ratio (possibly in the first section of the text) since it is one of the major parameters discussed

in this study. Overall, I am comfortable with the conclusions and support publication of this manuscript with minor edits, as detailed below.

Specific Comments:

Abstract The Abstract refers to 45 mangrove and 16 saltmarshes worldwide, but Fig. 1 and the Methods refer to 38 mangrove and 8 saltmarsh-dominated sites. Also, lines 144-145 refer to 33 mangrove and 10 saltmarsh sites. Please correct.

Lines 68-69 You describe alkalinity as mostly coming from HCO_3^- , and DIC as the sum of CO_2 , HCO_3^- and CO_3^{2-} . Are you assuming that the CO_3^{2-} concentrations in these systems are quite small, so that they do not contribute substantially to TA, therefore the major contributor to TA is HCO_3^- ? Are there other major contributors to TA other than HCO_3^- ? It should be clarified here why DIC causes acidification and TA causes more buffering. Is it because of the CO_2 component of DIC, which is not included in TA, since it is uncharged?

Line 84 (this paragraph): The ratio of TA:DIC is a major property of carbonate chemistry and is the major focus of this paper. However, it is not very clearly defined. Please explain here what it refers to, what it is used for, why it is used, etc. You mention in line 86 that it relates to the ratio of CO_3^{2-} to HCO_3^- , which is somewhat unclear since earlier on this page you define DIC as containing CO_3^{2-} , and HCO_3^- is TA. Again, there is confusion, since we know TA contains HCO_3^- and CO_3^{2-} ... (see previous comment). Some of this is mentioned in the Supplemental section (Figs. S11-13), but it still not completely clear. At a minimum, please refer to this supplemental material in this first section of the main text.

Figure 1: I am not aware of any saltmarshes located along the north coast of Alaska. I'm not sure why they are included in your map.

Lines 114-117 Given the TA:DIC slopes that different processes produce (aerobic respiration, sulfate reduction, etc.), how did you decide what contribution is made by the individual processes to the value measured at the sites? For example, 0.82 was the mean for porewater, which seemingly could have come from just one or a mix of all of the processes you listed. How is the assignment made as to which are important for each site? There was no description of this in the Methods.

Lines 117-120 You state that TA values of typical seawater are 2350 $\mu\text{mol}/\text{kg}$. Figure S1a does not support the statement that most mangrove and saltmarsh surface waters were well above this value. In fact, both mangrove SW and saltmarsh SW medians are below this value.

Fig. 2c The tidal amplitude circle sizes are not big enough. They are too difficult to discern, although the stated relationship between DIC and TA export and tidal amplitude is very cool! Also, the 0:0 grey lines are hard to see but useful. Please darken or make easier to see.

Line 140 How did you calculate the change in surface water pH during low vs high tides? I don't see this in Figure 2 or the Methods.

Lines 161-162 I'm unsure what this sentence is trying to say. It is unclear.

Fig. S9-S10 For panel f, what is meant by "unsigned latitude"? Also, for these figures, and regarding the relationships between TA, DIC and the potential environmental controls, it may be more appropriate to run a multivariate multiple regression analysis rather than several separate regressions.

Line 201 Insert a comma after "outwelling"

Methods Is the TA and DIC data presented? I only see the TA:DIC ratio in Table S2. It could possibly be in the electronically available dataset, but that is inaccessible at this time.

Also, please refer in the Methods and the first section of the main text to the supplemental section on the TA:DIC ratio (Figs. S11-S13), which I missed the first time I read through. This is very important information to bring attention to that helps with understanding the paper.

Reviewer #3 (Remarks to the Author):

First, I want to apologize to authors for being late with my review. I recently changed jobs (and changed computers and email accounts well) and at some point this review email invite fell through the cracks and I couldn't find it in any of my email accounts. I was glad when the editor reached out to me asking whether or not I was going to be able to return my review. Thanks for that, and again, apologies to everyone for having delayed this process.

This review provides an amazing summary on coastal wetlands biogeochemical processes that control the production and export of DIC and total alkalinity (TA) to coastal oceans and how TA:DIC ratios influence the coastal carbonate system and coastal ocean acidification.

The quantitative exploratory analyses presented in this manuscript really help setting up future site-specific hypothesis-testing driven studies and provides a roadmap for where new research focus should be placed on to fill spatial and temporal gaps to better constrain the contribution of TA and DIC to global coastal oceans carbon budgets.

This study does a great job explaining in detail and backed up by available published estimates the relevance of currently overlooked lateral TA exports as a long-term carbon sequestration mechanism, and its revised estimates improve our understanding towards better constraining the global coastal carbon budget.

Overall, this review is extremely well written and timely. I was really pleased to review this manuscript and am happy to recommend for publication in its current form. Such a great contribution to our field!

Here's a couple of tiny suggestions authors may want to consider if they end up revising this ms per other referee's suggestions:

1. On page 5, spell out 'DOC' since this is the first time that it appears in the text.
2. On Fig. 4, why not to show DIC and TA separately? Sorry if I missed something, but made me think it could be relevant since TA represents a long-term carbon sink.

Cheers!

Andre Rovai

Carbonate chemistry and carbon sequestration driven by inorganic carbon outwelling from mangroves and saltmarshes

Response to Reviewers

Reviewer 2

General Comments:

The authors present a study in which they examine DIC and TA concentrations, environmental controls, and outwelling in mangrove and saltmarsh systems. They assemble a global dataset of over 3000 porewater and surface water samples measured during timeseries and spatial surveys. The authors report that for the majority of their sites, more DIC than TA was exported to the coastal ocean, potentially enhancing coastal acidification. DIC outwelling was the primary (characterized) fate of primary production by both mangroves and saltmarshes--greater than CO₂ outgassing, burial, and organic carbon outwelling. Moreover the magnitude of DIC export from mangroves and saltmarshes combined was equal to ~15% of riverine-exported DIC, though this term is not yet accounted for in global carbon budgets.

This is a novel and interesting study that will be well received by the readers of this journal. The manuscript is well written, clear, and well organized, and I think this is a strong and comprehensive dataset. There are a lot of display items (4 figures + 13 supplemental figures and 6 supplemental tables), but they all seem to serve a purpose, so I don't recommend dropping any. The Methods are quite brief, which I realize is the norm for these short articles. However, they could include more information about the TA:DIC ratio (possibly in the first section of the text) since it is one of the major parameters discussed in this study. Overall, I am comfortable with the conclusions and support publication of this manuscript with minor edits, as detailed below.

Response: We are very grateful for the clear and comprehensive comments that help to improve our manuscript. We now added more information about the TA:DIC ratio - see comments below.

Specific Comments:

The Abstract refers to 45 mangrove and 16 saltmarshes worldwide, but Fig. 1 and the Methods refer to 38 mangrove and 8 saltmarsh-dominated sites. Also, lines 144-145 refer to 33 mangrove and 10 saltmarsh sites. Please correct.

Response: We agree this was confusing. The abstract combines conclusions for datasets with concentrations (TA:DIC ratios) and outwelling rates. Some of these sites overlap, but some do not. L144-145 referred only to evidence from TA:DIC ratios versus tidal ranges and estuarine mixing models. To prevent confusion and enhance precision, we added detail to those statements. First, we modified line 144-145 "Overall, tidal dynamics and mixing models reveal that 23 out of 33 mangrove sites and all saltmarsh sites (n = 10) had higher DIC than TA inputs potentially enhancing local acidification. Sites without water level data were excluded from the analysis." Second, we added to methods: "There was a partial intersection between the study sites where outwelling rates were measured (Fig. 3 and Supplementary Fig. 8) and the locations with TA and DIC observations (Fig. 1)."

Lines 68-69 You describe alkalinity as mostly coming from HCO₃⁻, and DIC as the sum of CO₂, HCO₃⁻ and CO₃²⁻. Are you assuming that the CO₃²⁻ concentrations in these systems are quite small, so that they do not contribute substantially to TA, therefore the major contributor to TA is HCO₃⁻? Are there other major contributors to TA other than HCO₃⁻? It should be clarified here why DIC causes acidification and TA causes more buffering. Is it because of the CO₂ component of DIC, which is not included in TA, since it is uncharged?

Response: Yes, it is because of the CO₂ component of DIC. We specified: “Mangroves and saltmarshes produce organic carbon that is partially mineralized, releasing inorganic carbon in the form of carbonate alkalinity (mostly HCO₃⁻ at pH<8) and dissolved inorganic carbon (DIC = CO₂ + HCO₃⁻ + CO₃²⁻). We also clarified: “Exported DIC, in contrast, can enhance coastal acidification by forming carbonic acid when CO₂ reacts with water and partly returns CO₂ to the atmosphere via air-sea exchange.”

Line 84 (this paragraph): The ratio of TA:DIC is a major property of carbonate chemistry and is the major focus of this paper. However, it is not very clearly defined. Please explain here what it refers to, what it is used for, why it is used, etc. You mention in line 86 that it relates to the ratio of CO₃²⁻ to HCO₃⁻, which is somewhat unclear since earlier on this page you define DIC as containing CO₃²⁻, and HCO₃⁻ is TA. Again, there is confusion, since we know TA contains HCO₃⁻ and CO₃²⁻... (see previous comment). Some of this is mentioned in the Supplemental section (Figs. S11-13), but it still not completely clear. At a minimum, please refer to this supplemental material in this first section of the main text.

Response: We feel that that we had made the narrative overly complicated in the introduction. We now modified the introduction: “The TA:DIC ratio relates to buffer factors and is thus a major property of carbonate chemistry determining the buffering capacity of seawater against external acid inputs¹⁸.” We also added “The use of TA:DIC ratios as a proxy for acidification or buffering is explained in detail in the Supplementary Methods.” to the main text after the first mention of TA:DIC ratios and cite Fig S11-S13 in the methods. For further reference we also added more citations to the supporting information: “Therefore, the TA:DIC ratio is widely used as a proxy of carbonate equilibrium and speciation in the context of ocean acidification²⁻⁵.”

Figure 1: I am not aware of any saltmarshes located along the north coast of Alaska. I’m not sure why they are included in your map.

Response: The orange background contour of Figure 1 showing saltmarsh occurrence in Alaska was obtained from previous global compilations. We have double checked Ref 19 as well as a recent saltmarsh review (Xin et al., 2021, Reviews of Geophysics). Both manuscripts show the occurrence of small saltmarsh patches in Alaska (see Figure 1 and Figure 12 in Xin et al. 2021). We feel that the Figure should remain with saltmarshes in Alaska. We notice that the orange background serves only to illustrate the broad ranges of mangroves and saltmarshes and has no direct influence in our general interpretation.

Lines 114-117 Given the TA:DIC slopes that different processes produce (aerobic respiration, sulfate reduction, etc.), how did you decide what contribution is made by the individual processes to the value measured at the sites? For example, 0.82 was the mean for porewater, which seemingly could have come from just one or a mix of all of the processes you listed. How is the assignment made as to which are important for each site? There was no description of this in the Methods.

Response: We agree with the reviewer’s perception and treat those values with care. We now address limitations in methods: “To determine dominant biogeochemical pathways for each site was beyond the scope of this study, since it requires analysis of respiration pathways (e.g., aerobic respiration, sulphate reduction etc.) and carbon isotopic signatures.” The processes mentioned in the main text refer to general trends documented in the literature.

Lines 117-120 You state that TA values of typical seawater are 2350umol/kg. Figure S1a does not support the statement that most mangrove and saltmarsh surface waters were well above this value. In fact, both mangrove SW and saltmarsh SW medians are below this value.

Response: We expanded the sentence to clarify: "...38% of mangrove and 35% of saltmarsh surface water samples exceeded typical seawater values of 2350 $\mu\text{mol/kg}$..."

Fig. 2c The tidal amplitude circle sizes are not big enough. They are too difficult to discern, although the stated relationship between DIC and TA export and tidal amplitude is very cool! Also, the 0:0 grey lines are hard to see but useful. Please darken or make easier to see.

Response: We refit the scales to improve the readability and increased the amplitude circle sizes. We also made the 0:0 lines darker.

Line 140 How did you calculate the change in surface water pH during low vs high tides? I don't see this in Figure 2 or the Methods.

Response: We added to methods: "To test the impact of tidal variation on the carbonate chemistry tidal ranges of DIC, TA, and pH were quantified. Tidal ranges were estimated from differences between DIC, TA, and pH values at low and at high tide. Minimum and maximum water levels were used to define low and high tides." We now added the tidal pH range per site to Supplementary Table 2 and refer to it in the main text.

Lines 161-162 I'm unsure what this sentence is trying to say. It is unclear.

Response: We modified to clarify: "Consequently, the increased buffer capacity of porewaters minimises the acidification potential caused by CO_2 release from intertidal wetlands."

Fig. S9-S10 For panel f, what is meant by "unsigned latitude"? Also, for these figures, and regarding the relationships between TA, DIC and the potential environmental controls, it may be more appropriate to run a multivariate multiple regression analysis rather than several separate regressions.

Response: We now clarify in caption: "Unsigned latitude refers to latitudes in both hemispheres disregarding the minus sign in the southern hemisphere to examine the impact of climate rather than hemisphere." We spent a considerable amount of time trying to build a multivariate multiple regression model to upscale fluxes globally. We agree with the reviewer that this would have been very valuable. However, no clear relationships between outwelling and environmental drivers emerged from those models. Hence, the existing datasets are not suitable yet to construct a significant model. We expect that this manuscript will lead to new research on the topic, including the construction of long term, detailed datasets that should enable the use of predictive models. As members of the growing community working on this topic, we feel that the field is at least 5 years away from having enough detailed datasets.

Line 201 Insert a comma after "outwelling"

Response: Correction made.

Methods Is the TA and DIC data presented? I only see the TA:DIC ratio in Table S2. It could possibly be in the electronically available dataset, but that is inaccessible at this time. Also, please refer in the Methods and the first section of the main text to the supplemental section on the TA:DIC ratio (Figs. S11-S13), which I missed the first time I read through. This is very important information to bring attention to that helps with understanding the paper.

Response: The DOI registration of the raw dataset is now completed and can be publicly accessed (<https://doi.org/10.1594/PANGAEA.949660>). We now added "The use of TA:DIC ratios as a proxy for acidification or buffering is explained in detail in the Supplementary Methods." to the main text after the first mention of TA:DIC ratios and cite Fig S11-S13 in the methods.

Reviewer 3

First, I want to apologize to authors for being late with my review. I recently changed jobs (and changed computers and email accounts well) and at some point this review email invite fell through the cracks and I couldn't find it in any of my email accounts. I was glad when the editor reached out to me asking whether or not I was going to be able to return my review. Thanks for that, and again, apologies to everyone for having delayed this process.

This review provides an amazing summary on coastal wetlands biogeochemical processes that control the production and export of DIC and total alkalinity (TA) to coastal oceans and how TA:DIC ratios influence the coastal carbonate system and coastal ocean acidification. The quantitative exploratory analyses presented in this manuscript really help setting up future site-specific hypothesis-testing driven studies and provides a roadmap for where new research focus should be placed on to fill spatial and temporal gaps to better constrain the contribution of TA and DIC to global coastal oceans carbon budgets.

This study does a great job explaining in detail and backed up by available published estimates the relevance of currently overlooked lateral TA exports as a long-term carbon sequestration mechanism, and its revised estimates improve our understanding towards better constraining the global coastal carbon budget.

Overall, this review is extremely well written and timely. I was really pleased to review this manuscript and am happy to recommend for publication in its current form. Such a great contribution to our field!

Response: We thank the reviewer for this encouraging and supportive feedback.

Here's a couple of tiny suggestions authors may want to consider if they end up revising this ms per other referee's suggestions:

1. On page 5, spell out 'DOC' since this is the first time that it appears in the text.
Response: Correction made.
2. On Fig. 4, why not to show DIC and TA separately? Sorry if I missed something, but made me think it could be relevant since TA represents a long-term carbon sink.
Response: We agree. We now show TA in Fig. 4 separately.

Cheers!

Andre Rovai

Once again, we thank the reviewers for their detailed feedback. We feel that those comments significantly improved our manuscript.

END OF REVIEWS